# Analysis of the Intestinal and Faecal Bacterial Microbiota of the Cervidae Family Using *16S* Next-Generation Sequencing: A Review

**DOI:** 10.3390/microorganisms11071860

**Published:** 2023-07-24

**Authors:** Irene Pacheco-Torres, David Hernández-Sánchez, Cristina García-De la Peña, Luis A. Tarango-Arámbula, María M. Crosby-Galván, Paulino Sánchez-Santillán

**Affiliations:** 1Livestock Program, Postgraduate College, Texcoco 56230, Mexico; pacheco.irene@colpos.mx (I.P.-T.); maria@colpos.mx (M.M.C.-G.); 2Faculty of Biological Sciences, Juarez University of the State of Durango, Gomez Palacio 35010, Mexico; cristina.garcia@ujed.mx; 3San Luis Potosi Campus, Postgraduate College, Salinas de Hidalgo 78620, Mexico; ltarango@colpos.mx; 4Faculty of Veterinary Medicine and Zootechnics No. 2, Autonomous University of Guerrero, Cuajinicuilapa 41940, Mexico; ssantillan@uagro.mx

**Keywords:** cervids, microbiota, pathogens, zoonotic bacteria, *16S* gene

## Abstract

The Cervidae family has a wide distribution due to its adaptation to numerous ecological environments, which allows it to develop a diverse microbial community in its digestive tract. Recently, research has focused on the taxonomic composition and functionality of the intestinal and faecal microbiota of different cervid species worldwide, as well as their microbial diversity and variation under different associated factors such as age, sex, diet, distribution, and seasonal variation. In addition, there is special interest in knowing how cervids act as reservoirs of zoonotic pathogenic microorganisms, which represent a threat to public health. This review provides a synthesis of the growing field of microbiota determination in cervids worldwide, focusing on intestinal and faecal samples using *16S* next-generation sequencing. It also documents factors influencing microbial diversity and composition, the microorganisms reported as pathogenic/zoonotic, and the perspectives regarding the conservation of these species. Knowing the interactions between bacteria and cervid health can drive management and conservation strategies for these species and help develop an understanding of their evolutionary history and the interaction with emerging disease-causing microorganisms.

## 1. Introduction

The intestinal microbiome is important in host development and health due to its association with nutrient acquisition, immune response, physiological functions, and behaviour [1,2,3]. For this reason, studies are focused on understanding the composition and functionality of animal intestinal microbiomes [4,5], which evolve together with their host [6].

Cervidae is the second most numerous family of artiodactyls worldwide, with a total of 46 species [7]; they occur from the tropics to the arctic regions, adapting to diverse environments [8]. The first study of gut microbiota in cervids was published by Li et al. [9], where massive sequencing of the *16S rRNA* gene was used to characterise the bacteria present in the gastrointestinal tract (GIT) of Chinese roe deer (*Capreolus pygargus*). In that study, variation in bacterial community composition was found across the GIT. Since then, several studies have used sequencing technologies to describe intestinal microbiota in cervids, generating knowledge on bacterial abundance and diversity [10,11,12,13]. However, several factors influence bacterial community composition, such as sex [14], diet [15], spatial distribution [16,17], and even seasonal variation [18,19]. Parallel to studies on GIT bacteria, various other studies have been conducted on the composition and function of ruminal microorganisms, especially bacteria, archaea, and protozoa [20,21,22,23,24,25,26]. The bacterial community composition in cervids is important in shaping the immune system and some chemical processes within the host; however, the community contains various pathogenic bacteria [15,27] considered of public health importance [28].

This review deals with (a) the current state of research on the faecal and intestinal bacterial microbiota of the Cervidae family worldwide, (b) factors influencing their composition, (c) bacteria involved in animal health with zoonotic incidence, and (d) future perspectives and implications for the conservation of these ruminant species.

## 2. Sources of Information

The articles referenced in this review were obtained from online scientific databases such as PubMed, Scopus, JStor, and ScienceDirect. Keywords such as “deer”, “Cervidae”, “microbiome”, “fecal and intestinal microbiota”, “zoonotic and pathogenic bacteria” and “*16S rRNA* gene” were used to search for articles related to this topic. The articles included in this review met the following inclusion criteria: (a) species of deer, (b) intestinal or faecal microbiota, (c) zoonotic and pathogenic bacteria in deer, and (d) use of sequencing techniques and the *16S* gene. The publication year was not limiting. The information considered included the deer species studied, localisation, wild or captive status, type of sample used, techniques used, the zoonotic or pathogenic potential of the host, and factors associated with the obtained results.

## 3. Faecal and Intestinal Microbiota and Microbiome in Cervids

The application of recent technologies, such as *16S rRNA* gene sequencing, allows us to study and obtain a better understanding of the diversity and composition of intestinal and faecal bacteria in the family Cervidae. These technologies have revealed a large number of unreported bacteria; however, current information is still limited as it is restricted to certain species in specific areas of the world like Asia, Europe, and North America. Therefore, there is still a wide field of ongoing research on the microorganisms associated with these deer species (Table 1, Figure 1).

The available information on intestinal and faecal microbiota shows that it is highly diverse. Li et al. [9] demonstrated the existence of a large number of bacteria associated with the GIT of Chinese roe deer (*Capreolus pygargus*). They reported 2223 OTUs (operational taxonomic units) assigned to 12 phyla and 87 genera and found that the bacterial diversity in the ileum, colon, and faeces was similar. In addition, the phyla Firmicutes and Bacteroidetes were reported to be the most abundant phyla in the GIT at 57% and 37%, respectively, coinciding with the findings for other species such as Svalbard reindeer (*Rangifer tarandus platyrhynchus*) [10], Sika deer (*Cervus nippon*) [11], white-lipped deer (*Cervus albirostris*) [32], Pére David’s deer (*Elaphurus davidianus*) [12], red deer (*Cervus elaphus*) [34,36], fallow deer (*Dama dama*) [34], and white-tailed deer (*Odocoileus virginianus*) [14,19]. The importance of these phyla lies in food fermentation [10,31], and they are associated with body size and fat reserves in individuals [43,44].

A phylosymbiotic evaluation of the faecal microbiota from seven species of captive-bred cervids (*Axis porcinus*, *Cervus elaphus*, *C. nippon*, *Rusa unicolor*, *Dama dama*, *Elaphurus davinianus*, and *Elaphodus cephalophus*) resulted in the identification of 8849 OTUs, where the phyla Firmicutes, Bacteroidetes and Spirochaetes and the family Ruminococcaceae predominated in most samples [30]. In Svalbard reindeer (*Rafinger tarandus platyrhynchus*), a dominance of the phyla Firmicutes and Bacteroidetes, covering 95% of the identified bacterial sequences, was reported [10]. In white-tailed deer (*Odocoileus virginianus*), Firmicutes were more abundant in winter than in summer. However, the abundance of the phylum Bacteroidetes did not vary between seasons [16]. In farmed Sika deer (*Cervus nippon*), Bacteroidetes were more abundant than Firmicutes [11]. For white-lipped deer (*Cervus albirostris*), the microbiota from free-living and captive populations was characterised, and 33 phyla, 67 classes, 172 orders, 305 families, and 637 genera were classified. Of this classification, 17 phyla, 27 classes, 77 orders, 133 families, 261 genera, and 309 species were shared between both groups. Firmicutes and Bacteroidetes were abundant (>84%) in the samples. In free-living populations, Firmicutes were considerably abundant (63% to 82%) [13]. In Père David’s deer or Milú, 12 phyla, 22 classes, 27 orders, 47 families, and 49 genera were classified, with the phyla Firmicutes and Bacteroidetes being the most abundant [12]. In captive red and fallow deer, 20 and 18 phyla were recorded, respectively, where Firmicutes and Bacteroidetes showed an abundance greater than 70% of the total sequences in the samples. In addition, 89 families and 193 genera were reported in red deer samples, whereas 102 families and 227 genera were reported in fallow deer; in both species, the 10 most abundant genera were similar [34].

Ishida-Kuroki et al. [39] reported the abundance of taxa from the family Ruminococcaceae in free-living Sika deer (*C. nippon nippon* and *C. nippon aplodontus*). In white-lipped deer, the families Clostridicaceae, Ruminococcoccaceae, Prevotellaceae, Bacteroidaceae, Lachnospiraceae, Rikenellaceae, and Christensenellaceae were the core taxa [13].

Delgado et al. [16], in a study on the family Ruminococcaceae, reported the presence of an unclassified genus as predominant in white-tailed deer samples, whereas species of the genus *Ruminococcus* were described as predominant in Svalbard reindeer [10]. The genus *Monoglobus* was reported in white-lipped deer [13]. Minich et al. [14] reported the association between Firmicutes and Bacteroidetes as providing a higher energy extraction and better efficiency in diet fermentation in addition to participating in fat accumulation for winter survival [19]. In free-living deer, the microbiota can maximise the energy gained with a fibrous diet from browsing, whereas grain-rich diets fed to captive deer reduce the need for fermentation efficiency and create a niche for microbial taxa capable of metabolising starches and soluble sugars [14]. The species *Ruminococcus flavefaciens*, *R. albus*, and *Fibrobacter succinogenes* contribute to fibre degradation [10,11,16] in herbivores and are therefore not exclusive to cervids [45].

Although the sequence classification techniques and databases used in various studies tend to vary due to the specific considerations of each author, there is consistency among the results on bacterial diversity obtained for different cervid species documented to date. These results are closely related to phyllosymbiosis, i.e., the association among faecal microbiota with respect to the host species; this allows for generating a guideline on the coevolution of cervid species and their microbiota [30,46,47].

## 4. Factors Influencing the Composition of Faecal and Intestinal Microbiota in Cervids

Advances in molecular biology, such as next-generation sequencing, make it possible to specifically address the association between a host and its microbiota. In animals, the microbiota can take various forms, from symbiotic associations to the presence of pathogenic microorganisms [48]. Diverse factors affect the microbiome composition of animals such as age, sex, feeding, distribution, seasonal variation, social behaviour, and captivity [13,39,49].

Several studies document the influence of captivity on the composition of the faecal microbiota in animals, resulting in reduced proportions of microbial diversity in relation to free-living populations. This is associated with extreme modifications, mainly due to changes in feeding, restricted habitats, reduced social interactions, and high exposure to antibiotics [50,51,52,53,54].

These alterations in the microbiota caused by captivity have a negative impact on the health and reproductive performance of cervid populations. They also affect the survival of individuals reintroduced into the wild if the microbiota compromises host digestive or immune functions in the environment [50,55,56]. Carthey et al. [57] indicated that these alterations are related to the loss of microbial communities associated with a species, which will have an impact on individuals born under these conditions, resulting in the gradual loss of diversity and reduced opportunities to acquire a diverse set of microbes.

However, captivity is often a necessary resource for the conservation of species at risk of extinction; therefore, further research is required regarding the composition of cervid microbiota associated with changes influenced by captive breeding and diet formulation, all of which can impact microbial diversity. Possibly, the role of captivity in conservation can be expanded and included in future management practices [56].

The association between captivity and gut microbiota in cervids was studied extensively, comparing captive and free-living populations. In some cases, the objective was to generate a baseline of knowledge for the conservation of wild populations [31].

### 4.1. Diet-Associated Deer Microbiota

Habitual diets provide a consistent source of dietary substrates to the microbiota, creating an environment that continuously keeps microbial ecology without influencing the healthy core microbiota [42,58]. However, the feeding type is a key factor in compositional changes in these communities [59,60,61].

Principal changes in a diet type are associated with the captivity of deer populations. This action modifies feeding behaviour, causing an unfavourable effect on cervid populations by exerting selective pressure on microorganisms and resulting in a higher probability of encountering opportunistic pathogens in these populations [13,18,39].

In Rocky Mountain elk *Cervus canadensis nelsoni*, feeding with supplementation using alfalfa pellets generated changes in the composition of the bacterial microbiota, in contrast to supplementation using hay [15]. In Pére David’s deer, Sun et al. [41] reported that differences in diets did not have an impact on the diversity and richness of the bacteria microbiome; however, the differences had an effect on the microbial community structure, principally causing changes at genera level [41]. Another study found that deer fed with a combination of silage and natural vegetation had a lower diversity in contrast to those fed a regular diet composed entirely of silage or natural vegetation, which favoured a greater diversity of intestinal microorganisms in the host [42].

In free-living and captive Siberian roe deer, the abundance of Bacteroidetes and Firmicutes is associated with diet type [31], while, in red deer in enclosures, the proportion of Firmicutes is related to silage consumption [18]. It was found that the most representative taxonomic group in captive populations of deer was the phylum Bacteroidetes, whereas Firmicutes played an important role in free-living populations [21,22]. In captivity, these phyla are associated with unhealthy diets [62]. The abundance of these phyla and differences in the composition and biological function of the gut microbiome associated with diet in free-living and captive populations were reported for white-lipped deer [13], white-tailed deer [14], red deer [35,36,37], and Sika deer *Cervus nippon* [11,63]. Free-living populations present a greater microbial diversity due to greater access to different types of food available [63]. The industrial food fed long-term to captive populations affects and shapes the gut microbes, alters functions of the gut microbiota, and generates unhealthy conditions in the host [13]. A higher portion of the genera *Bacteroides* and *Prevotella* was found in captive populations, which was related to the influence of the supplied diet and rearing conditions, propitiating a markedly different gut microbiota in white-tailed deer [14].

### 4.2. Deer Microbiota Associated with Distribution and Seasonal Variation

Intestinal microbial communities are complex, dynamic, spatial, and time-based. These communities present differences in their composition among individuals that inhabit different environments [16,64,65]. Seasonal changes and spatial locations influence variation in the vegetation available for feeding, and this influences the diversity and composition of the faecal microbiota [16,19]. Nutrient procurement is a fundamental challenge for wild animals as the availability and nutritional content of feed can vary temporally and spatially in response to changes in climate and geography. This variation leads to a change in the host gut microbiota [17,66,67,68]. However, there are core bacterial communities that coincide among individuals of the same species that have remained for a long time within a host and are identified at any season of the year [65].

The gut microbiota in cervids is associated with spatial location and seasonal changes [19]. Delgado et al. [16] found a correlation between spatial location and faecal microbiota composition in white-tailed deer. In addition, space, habitat used, and seasonal changes were also found to be correlated with the gut microbiome in this deer [19]. In North American elk (*Alces alces*), gut microbial communities associated with the co-occurrence of individuals and interspecific associations were reported [29]. Menke et al. [18] reported that the abundance of gut microbiota in red deer inhabiting enclosures was related to the distribution of the sampled hibernation sites. In Pére David’s deer, a difference of 88 bacterial taxa at the phylum and genus levels was shown, which was attributed to changes in the populations’ environment and feeding patterns [12]. In two mule deer (*O. hemionus*) populations from Cache County and Monroe Valley in Utah, USA, differences associated with microbial abundance, geography, and seasonal changes were reported. There was mainly an increase in the abundance of Coriobacteriales in the Cache population, which was related to the fat content in deer [17].

Studies on gut microbiota composition showed that the alpha diversity of bacterial communities was higher in the summer than in the winter, which was related to a decrease in the availability of feed for mule deer [17] and white-tailed deer [19]. In white-lipped deer (*Cervus albirostris*), the phyla Firmicutes, Bacteroidetes, and Patescibacteria were enriched during the herbaceous season, whereas the abundances of Actinobacteria and Proteobacteria were increased in the wilting season [33].

### 4.3. Age- and Sex-Associated Microbiota in Deer

Several studies point to the existence of a correlation between age, sex, and gut microbiota diversity [69,70,71,72]. In Sika deer, age is indicated as a key factor in microbial succession, resulting in increased richness and diversity of the microbiota [38]. In captive white-tailed deer, no difference was detected between males and females [14,16]; however, in free-living deer, differentially abundant taxa were found by sex. *Oscillobacter* and bacteria of the non-culturable genus GCA-900066575, belonging to the family Lachnospiraceae, were significantly increased in males [14]. A study on wild red deer showed that bacterial diversity was higher in females than in males, which was attributed to hormones [36]. Moreover, in a captive population, the diversity and composition of the gut microbiota were different at different growth stages [37].

### 4.4. Cervid Microbiota Associated with Antibiotics Treatments

Intestinal bacterial communities are related to host health [41,73,74,75,76]. According to Lange et al. [77], antibiotics modify the composition and function of the microbiota. The use of antibiotic treatments implies a decreased microbial diversity and increased colonisation by invading pathogens [78]. Hu et al. [79] pointed out that an anthelmintic treatment in Sika deer modified bacterial communities in terms of alpha diversity and caused a reduction in the genus Bacteroides because of side effects [79]. However, the effects of antibiotic treatments need to be studied further in cervids, principally in captive populations due to sanitary management.

### 4.5. Deer Microbiota Associated with Interspecies Variation

Another factor associated with a difference in microbiota composition is related to interspecies variation. In the literature, differences in the abundance of bacterial communities at the phylum level are reported among species because of a possible relationship with gut physiological characteristics [30,34,80]. This relationship was found in captive red and fallow deer, and differences in species richness and abundance were also found [34]. Indeed, these differences were also reported for other deer species of the Cervidae family [30]. However, this association is scarcely documented, so it is important and necessary to complete more investigations.

## 5. Pathogenic and Zoonotic Bacteria in Faecal and Intestinal Samples from Cervids

Most animals harbour a high portion of pathogens in their intestinal microbiota, making them important reservoirs of zoonotic pathogens and posing certain risks to public health [39]. A balanced bacterial microbiota is important for host health [81]; however, in a state of dysbiosis, it causes an increase in the abundance of potential pathogenic microorganisms, with negative effects on health [82,83,84].

Studies on deer provide information on emerging pathogens and the detection of zoonoses due to their distribution, abundance, and behaviour [28]. Deer can become infected with pathogenic microorganisms by eating or drinking contaminated material, thus becoming carriers of bacteria that spread in their environment, which makes them a link in the occurrence of zoonotic diseases [27,85,86].

In cervids, *Escherichia coli*, *Yersinia enterolitica*, *Y. ruckerii*, *Aeromonas sobria*, *Enterococcus faecium*, *Staphylococcus aureus*, and *Lysteria monocytogenes* are reported as potential health threats [18,28]. In one study, a high abundance of *E. coli* was detected in red deer in winter enclosures, indicating an animal–human relationship [18]. This bacterial species has also been reported in red deer [87,88,89], roe and fallow deer [88], and Sika deer (*Cervus nippon*) [39].

In New Zealand-farmed red deer, *Campylobacter* spp., *E. coli*, *Enterococcus* spp., and *Yersinia* spp. were also detected [90]. Mackintosh et al. [91] identified bacteria such as *Bacillus anthracis*, *Brucella* spp., *Clostridium perfringens*, *Mycobacterium avium paratuberculosis*, *Leptospira interrogans*, *Fusobacterium necrophorum*, *Pasteurella multocida*, *Salmonella* spp., *Mycobacterium bovis*, and *Yersinia pseudotuberculosis* as causes of disease in populations of *Cervus elaphus*, *Alces alces*, *Dama dama*, *Odocoileus virginianus*, *Ranfinger tarandus*, *Axis axis*, and *Cervus unicolor*.

Ghielmetti et al. [27] described *Mycobacterium microti* in red deer individuals from Austria and Germany as a species of veterinary importance due to its pathogenicity in wildlife and its zoonotic potential. Direct transmission of this bacterium between individuals is usually unlikely, and transmission may therefore be related to the ingestion of food or water from contaminated sources. In Sika deer, 29 pathogens were identified with cumulative relative abundances ranging from 2.3% to 39.9%; most of them were *E. coli-Shigella*. These bacterial species were positively correlated with other species that have pathogenic potential, such as *Salmonella enterica*, *Campylobacter jejuni*, and *Klebsiella pneumoniae* [39].

Wildlife plays an important role in maintaining ecological balance and biodiversity. Animal health has a considerable influence on human safety, given the large number of unknown microorganisms that can be carried by wildlife [92].

In recent years, outbreaks of emerging diseases have occurred more frequently, affecting wildlife and generating zoonoses [93]. The emergence of these infectious pathogenic microorganisms is associated with multiple factors such as anthropogenic impact, climate change, biodiversity loss, habitat degradation, and the increasing rate of wildlife–human contact [92,93,94]. Wildlife is considered an important source of microorganisms that cause infectious diseases [95] and an important carrier of zoonotic pathogens [28], making it necessary to focus on the composition of the faecal and intestinal microbiota of cervids to establish a clearer picture of the presence of bacteria with possible pathogenic and/or zoonotic potential that can be found in different species throughout the world.

## 6. Perspectives on Microbiome and Microbiota Research in Cervids

Research on the characterisation of the intestinal and faecal bacterial microbiota associated with wild species has become important in the effort to understand the interaction between microorganisms and a host [96]. However, most research focuses on cataloguing the composition of bacterial communities, rarely considering the influence of these on the adaptive potential of wild species [97]. In cervids, information is scarce and limited to specific species (Figure 2); however, it allows a better understanding of the structure of the microbiota composition in these ruminants. Diet is one of the main factors that interfere with microbial community composition in both wild and captive animals [15]. Likewise, the phyla Firmicutes and Bacteroidetes predominate in bacterial communities [31].

The improvement in knowledge on the microbiota–host association paved the way for a new area of research focused on the conservation biology of endangered species and populations affected by anthropogenic disturbances [49,58,96,98].

From these studies, new questions arise, such as the importance of sequencing depth, bioinformatics analyses, and the quality of the databases used to identify bacterial community composition. The limitations of these considerations may have negative effects on the ability to identify and predict the functional role of bacteria and their interaction with their host, which plays a crucial role in shaping the metabolic and regulatory networks that define health and disease states in animals [1,96].

Microbiota research can be beneficial to conservation management programmes for cervid populations worldwide. There is a need to broaden the strategy by involving all factors that might be associated with bacterial community composition, given the poor understanding of animal microbial diversity and function [99].

## 7. Conclusions

The intestinal microbiota in cervids influences the metabolism, physiology, and development of the host immune system. Diet, age, distribution, seasonal changes, and captivity influence variation in the microbiota. It is necessary to deepen knowledge regarding the alteration caused by these factors, considering that cervid species as reservoirs for microorganisms with pathogenic and zoonotic potential that play an important role in the development of various diseases. Such information could be a milestone for a better understanding of the structural dynamics of bacterial communities in cervids, the evolutionary history, and the interaction between the host and microorganisms causing emerging diseases of veterinary and public health importance.

## Figures and Tables

**Figure 1 microorganisms-11-01860-f001:**
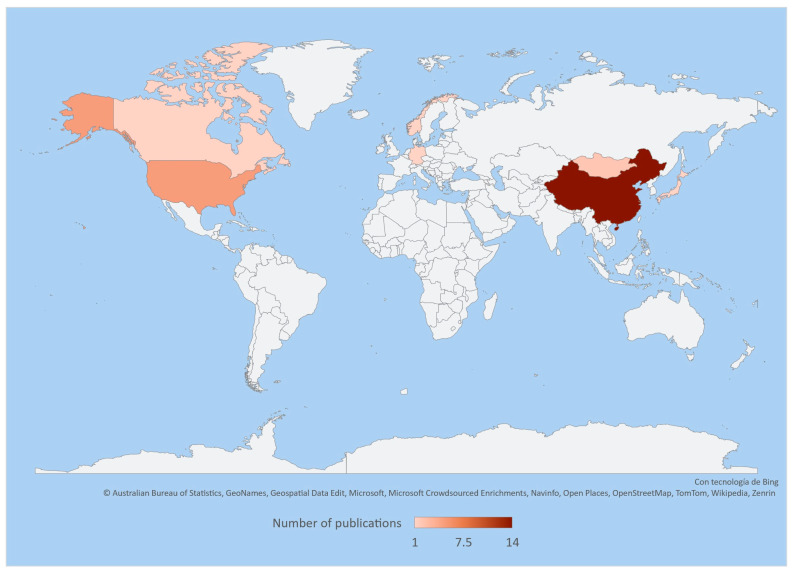
Heatmap representing the number of publications on the intestinal and faecal microbiota of cervids throughout the world based on the information and references in Table 1.

**Figure 2 microorganisms-11-01860-f002:**
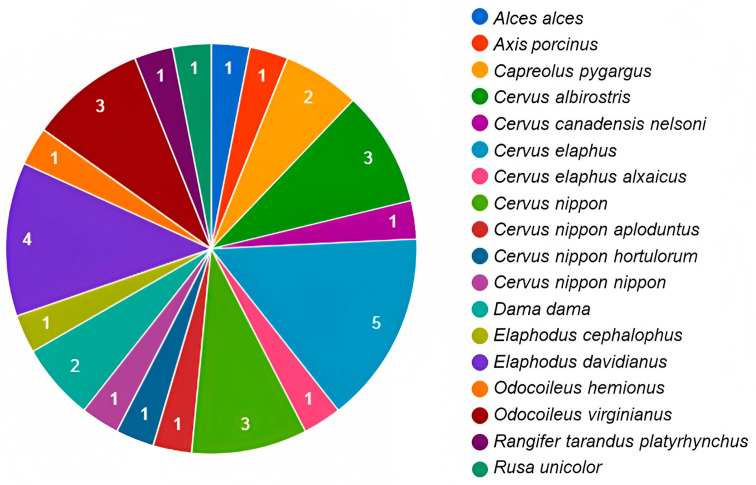
Number of publications on microbiota by species of cervids throughout the world. The numbers inside the graph are related to the amount of publications available for each deer species (elaboration based on the information and references in Table 1).

**Table 1 microorganisms-11-01860-t001:** Studies on the intestinal and faecal microbiota of different deer species.

Species	Origen Captive/Wild	Samples	Country Location	Target Gene	Approach	Reference
*Alces alces*	Wild	Faeces	USA	V4 *16S rRNA*	Illumina MiSeq	[29]
*Axis porcinus*	Captive	Faeces	China	V4-V5 *16S rRNA*	Illumina MiSeq	[30]
*Capreolus pygargus*	Wild	Ileum, colon, and faeces	China	V3-V4 *16S rRNA*	Illumina MiSeq	[9]
*C. pygargus*	Captive and wild	Faeces	China	V3-V4 *16S rRNA*	Illumina MiSeq	[31]
*Cervus albirostris*	Wild	Faeces	China	V3-V4 *16S rRNA*	Illumina MiSeq	[32]
*C. albirostris*	Captive and wild	Faeces	China	V3-V4 *16S rRNA*	Illumina NovaSeq	[13]
*C. albirostris*	Wild	Faeces	China	V3-V4 *16S rRNA*	Illumina MiSeq 2500	[33]
*C. canadensis nelsoni*	Wild	Faeces	USA	V4 *16S rRNA*	Illumina MiSeq	[15]
*C. elaphus*	Captive	Faeces	China	V4-V5 *16S rRNA*	Illumina MiSeq	[30]
*C. elaphus*	Wild	Faeces	Germany	*16S rRNA*	Illumina MiSeq	[18]
*C. elaphus*	Captive	Faeces	China	V3-V4 *16S rRNA*	Illumina MiSeq	[34]
*C. elaphus*	Captive and wild	Faeces	Mongolia	V3-V4 *16S rRNA*	Illumina HiSeq 2500	[35]
*C. elaphus*	Captive and wild	Faeces	Mongolia	V3-V4 *16S rRNA*	Illumina NovaSeq 6000	[36]
*C. elaphus alxaicus*	Captive	Faeces	China	V3-V4 *16S rRNA*	Illumina PE MiSeq	[37]
*C. nippon*	Captive	Faeces	China	V4-V5 *16S rRNA*	Illumina MiSeq	[30]
*C. nippon*	Captive	Jejunum and ileum	China	V3-V5 *16S rRNA*	Illumina PE MiSeq 250	[38]
*C. nippon*	Captive	Faeces	China	*16S rDNA*	Oxfor Nanopore MinION Mk1C	[39]
*C. nippon aploduntus*	Wild	Rectal faeces	Japan	V3-V4 *16S rRNA*	Illumina MiSeq	[40]
*C. nippon hortulorum*	Captive and wild	Faeces	China	V3-V4 *16S rRNA*	Illumina HiSeq 2500	[11]
*C. nippon nippon*	Wild	Rectal faeces	Japan	V3-V4 *16S rRNA*	Illumina MiSeq	[40]
*Dama dama*	Captive	Faeces	China	V4-V5 *16S rRNA*	Illumina MiSeq	[30]
*D. dama*	Captive	Faeces	China	V3-V4 *16S rRNA*	Illumina MiSeq	[34]
*Elaphodus cephalophus*	Captive	Faeces	China	V4-V5 *16S rRNA*	Illumina MiSeq	[30]
*E. davidianus*	Wild	Faeces	China	V3-V4 *16S rDNA*	Illumina MiSeq	[12]
*E. davidianus*	Captive	Faeces	China	V4-V5 *16S rRNA*	Illumina MiSeq	[30]
*E. davidianus*	Captive and wild	Faeces	China	V4-V5 *16S rRNA*	Illumina MiSeq	[41]
*E. davidianus*	Wild	Faeces	China	V3-V4 *16S rRNA*	NovaSeq6000	[42]
*Odocoileus hemionus*	Wild	Faeces	USA	V4 *16S rRNA*	Illumina HiSeq 2500	[17]
*O. virginianus*	Wild	Faeces	USA	V5-V3 *16S rRNA*	Roche 454 GS Junior	[16]
*O. virginianus*	Captive and wild	Faeces	USA	V4 *16S rRNA*	Illumina MiSeq	[14]
*O. virginianus*	Wild	Faeces	Canada	V3-V4 *16S rRNA*	Illumina MiSeq	[19]
*Rangifer tarandus platyrhynchus*	Wild	Faeces	Norway	V3-V4 *16S rRNA*	Illumina MiSeq	[10]
*Rusa unicolor*	Captive	Faeces	China	V4-V5 *16S rRNA*	Illumina MiSeq	[30]

## Data Availability

No new data were created or analysed in this study. Data sharing is not applicable to this article.

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
