# Peer review of "Analysis of the Intestinal and Faecal Bacterial Microbiota of the Cervidae Family Using 16S Next-Generation Sequencing: A Review"

_microorganisms, 2023, doi:10.3390/microorganisms11071860_

Round 1

Reviewer 1 Report

This article is a review about Cervidae family (deers) のintestinal and fecal microbiota. Interaction between the host and its intestinal microbiome recently attracts researchers to understand their evolutionary history and other biological concerns. The aim of this article might make attention in researchers. Structure of this article seems to be organized well. However, this article needs to be improved as follows.

L40: The authors abbreviated “the gastrointestinal tract” as TGI but this should be “GIT” as used in other references.

L61-62: The authors described “… current information is still limited as it is restricted to certain species in specific areas (Figure 1)”. The authors should show some examples of species and areas. The numbers in figure 1 are wrong: 0 should be 1 to 4, 5 should be 5 to 9, and so on.

L62-63: The authors described “… there is … with these deer species (Table 1)”. Order of species in Table 1 seems to be at random. The purpose of this review is to show microbiome in Cervidae so that table 1 should be reorganized using genus and species order.

L167-186: The authors cite references in L170-184 that the microbiome associates with diet. On the other hand, they cite a reference, 42, that there is no difference in L184-186. The reason for this lack of difference is coevolution of the speceis with its microbiota. If so, it is necessary to discuss why coevolution is not seen in some cases and not in others as a review.

Sections 3.1 and 3.6 are related to each other, so they should be in consecutive order.

Section 3.3 is a description of interspecies variation, but the other sections are intraspecies variations, so 3.3 should be placed after (at the end of) the other sections.

Section 3.5 should be about "space-time dynamics," but the content is different. It is necessary to describe what kind of phenomena the authors mean by "space-time dynamics" before reviewing.

Section 3.6 is a paragraph about deer microbiota associated with seasonal variation. What is different from seasonal variation and space-time dyamincs of Section 3.5. “Seasonal variation” might be related to “space-time dynamics”. The authors should describe what kind of seasonal variation affect deer microbiota clearly.

L255-292: The authors reviewed captive-associated microbiota of cervids. Does it mean that other sections, 3.1 to 3.5 and 3.8 are reviewed on free-living cervids? The authors should review microbiota of captive and free-living cervids carefully and clearly.

L315: The authors cited a reference 97 which is difficult to access for general readers. It is better to cite other references which are accessible for general readers.

Reviewer 2 Report

microorganisms-2450361

The issue under review of Pacheo-Torres et al. is very important, interesting and really constitutes a scientific niche, as relatively little work has been undertaken on this topic so far.

Interestingly, these animals are more and more often taken into account in the global crisis related to the lack of food, so from a technological point of view, breeding these animals is becoming more and more popular in Europe, and looking at the presented works, the topics related to microbiota and the safety of food from these species seem to be does not exist in the European area and is still developing at a global scale.

The work seems to be prepared carefully and comprehensively presents the discussed topic. The prepared graphics and tables have been prepared carefully and legibly. They fully meet the requirements of the journal. I can't fully read Figure 2 because it would be useful to use numerical values on a pie chart, but otherwise, the publication is clear and seems legible.

The only major doubt is the lack of a methodical description of the method of selecting the publications listed in this review. I suggest adding a subchapter in which the method of selecting works will be described, including the names of the searched databases, words used for searching, the range of years and additional search criteria used. This can help in assessing and tracking the correctness of the selection of works.

Minor Suggestions:

- removing the italics in the word Captive in the table in publication no. [38]

- adding numerical description to the pie chart (Fig.2) or remove Fig 2.

I have no bigger doubts about english. 

Round 2

Reviewer 1 Report

The authors modified the manuscript appropriately. The manuscript will provide better understanding on the microbiome of Cervidae to readers.

Reviewer 2 Report

It seems that the manuscript after the correction gained for its quality and the misleading parts were eliminated. I believe in this form it can be published due to the fact that this topic is a niche. 

No doubts about Quality